# Genome-Wide Association Study for Screening and Identifying Potential Shin Color Loci in Ducks

**DOI:** 10.3390/genes13081391

**Published:** 2022-08-04

**Authors:** Qixin Guo, Yong Jiang, Zhixiu Wang, Yulin Bi, Guohong Chen, Hao Bai, Guobin Chang

**Affiliations:** 1College of Animal Science and Technology, Yangzhou University, Yangzhou 225009, China; 2Joint International Research Laboratory of Agriculture and Agri-Product Safety, The Ministry of Education of China, Institutes of Agricultural Science and Technology Development, Yangzhou University, Yangzhou 225009, China

**Keywords:** duck, shin color, GWAS, melanin

## Abstract

Shin color diversity is a widespread phenomenon in birds. In this study, ducks were assessed to identify candidate genes for yellow, black, and spotted tibiae. For this purpose, we performed whole-genome resequencing of an F_2_ population consisting of 275 ducks crossed between Runzhou crested-white ducks and Cherry Valley ducks. We obtained 12.6 Mb of single nucleotide polymorphism (SNP) data, and the three shin colors were subsequently genotyped. Genome-wide association studies (GWASs) were performed to identify candidate and potential SNPs for the three shin colors. According to the results, 2947 and 3451 significant SNPs were associated with black and yellow shins, respectively, and six potential SNPs were associated with spotted shins. Based on the SNP annotations, the *MITF*, *EDNRB2*, POU family members, and the SLC superfamily were the candidate genes regulating pigmentation. In addition, the isoforms of *EDNRB2*, *TYR*, *TYRP1*, and MITF-M were significantly different between the black and yellow tibiae. *MITF* and *EDNRB2* may have synergistic roles in the regulation of melanin synthesis, and their mutations may lead to phenotypic differences in the melanin deposition between individuals. This study provides new insights into the genetic factors that may influence tibia color diversity in birds.

## 1. Introduction

Skin color diversity is a common phenomenon in most birds, and the color of the feathers, coats, and skin is mainly regulated by melanin and carotenoids. Melanin is primarily synthesized in melanocytes [1,2,3,4,5]. Previous studies have shown that skin color is highly heritable, one of the most variable phenotypic features, and influenced by both genetic factors and the environment [6,7]. The skin pigmentation is correlated with latitude, which could be due to the effect of UV light on pigmentation [8]. The skin pigmentation is also synergistically influenced by the effects of natural selection associated with climate, lifestyle, diet, and metabolism [9,10].

With rapid advances in genetics and genomics, scientists have realized that the diversity of human skin color is due to the natural positive selection of genes that influence human pigmentation, particularly genes in the melanosome biogenesis or melanin biosynthesis pathways [11,12,13]. Recently, a large number of polymorphisms in the genes associated with melanogenesis, such as *TYR*, *MCIR*, *BCOD2*, *IRF4*, *SLC45A2*, *KITLG*, *MITF*, and *OCA2*, have been shown to be associated with human skin color [4,14,15]. SAL suppresses skin inflammation and melanogenesis by targeting P4HB and regulating the formation of the IRF1/USF1 transcriptional complex [16,17,18]. In addition, SAL-plus may be a novel inhibitor of melanogenesis and inflammation [19]. The C-type lectin receptor, *CLEC12B*, is highly expressed in melanocytes, and its expression is reduced in dark skin compared to white skin. *CLEC12B* directly recruits and activates SHP1 and SHP2 through its immunoreceptor tyrosine-based inhibitory motif structural domain and promotes CRE *CLEC12B*, thereby ultimately controlling melanin production and hyperpigmentation in vitro and in reconstructed human epidermal models. The identification of *CLEC12B* in the melanocytes suggests that the C-type lectin receptor is involved in functions other than immunity and inflammation, and also provides insights into melanocyte biology and melanogenesis regulation [20].

As the second-largest poultry species in the world, ducks mostly have yellow and black shins. Spotting occasionally occurs, and this phenotype directly affects the marketability of carcasses. However, the genetic factors responsible for the appearance of a spotted color remains unclear. Thus, we conducted genome-wide association studies (GWASs) on 275 ducks and crossed F2 generations of Cherry Valley and Runzhou crested-white ducks to gain insights into the influence of genetic factors on shin pigmentation. Our findings provide insights into the molecular regulatory mechanisms underlying skin color and the genetic modifications that lead to the melanin deposition in duck shins.

## 2. Materials and Methods

### 2.1. Ethical Approval

All of the duck experiments were performed in accordance with the Regulations on the Administration of Experimental Animals issued by the Ministry of Science and Technology (Beijing, China) in 1988 (last modified in 2001). The experimental protocols were approved by the Animal Care and Use Committee of the Yangzhou University (YZUDWSY2017-11-07). All efforts were made to minimize animal discomfort and suffering.

### 2.2. Samples and Sequencing

The F_2_ resource population, which represented a cross between the Chinese crested (CC) and Cherry Valley ducks (CV), was obtained from the Laboratory of Poultry Genetic Resources Evaluation and Germplasm Utilization at Yangzhou University. The ducks were raised in stair-step cages under recommended environmental and nutritional conditions at the conservation farm of the Ecolovo Group, China. The CC duck has black shins and white plumage and represents an indigenous Chinese breed. The CV duck has yellow shins and white plumage and represents a commercial breed. In the F_1_ generation, 30 CC ducks and six CV ducks were randomly selected and divided into six families to interbreed. To generate F_2_ progeny, 30 males and 150 non-related female ducks were used as the hybrids. A total of 275 ducks were used in the re-sequencing and subsequent experiment. To identify the candidate genes associated with shin color, we classified the shin colors as yellow, spotted, or black (Appendix A). The ggcor package of R was used to calculate the correlation between shin color and sex.

Blood samples were used to collect high-quality DNA at 42 days of age. The genomic DNA (gDNA) was extracted from the blood samples using a DNA extraction kit (QIAampR DNA Blood Mini Kit), following the manufacturer’s protocol. Two paired-end sequencing libraries with insert sizes of 350 bp were constructed, according to the Illumina protocol (Illumina, San Diego, CA, USA). All of the libraries were sequenced on the Illumina NovaSeq platform.

### 2.3. Genotyping

The raw reads were filtered using the NGS QC Toolkit (version 2.3) with the default parameters. The clean reads were mapped to the duck reference genome (CC duck genome assembled in our lab (unpublished)), and a Burrows–Wheeler alignment (BWA aln) was performed using the default parameters [21]. The GATK was then used to perform single nucleotide polymorphism (SNP) in accordance with [22]. The VCF tools were used to further filter the raw data [23]. The SNPs were screened for a minor allele frequency (MAF) > 0.05, maximum allele frequency < 0.99, and maximum missing rate < 0.01. After filtering, the SNPs showed a mean density of 8.5 SNPs/kb across the genome. All of the filtered SNPs were distributed on 37 autosomal chromosomes, ChrZ and ChrU (unplaced scaffolds).

### 2.4. Population Structure

The population structure was assessed by multidimensional scaling (MDS) using the PLINK 1.9 software [24]. The independent SNPs were obtained on all of the autosomes using the in-dep-pairwise option with a 50 bp window, five steps, and an r^2^ threshold of 0.2. Pairwise identity-by-state (IBS) distances between all of the individuals were calculated using these independent SNP markers, and the MDS components were acquired using the mds-plot option based on the IBS matrix. A relative kinship matrix was constructed using the independent SNP markers. In addition, we used the trimmed SNP data and performed a dimensionality reduction using three methods (PCA, Uniform Manifold Approximation and Projection (UMAP), and PCA-UMAP). First, we performed a PCA analysis of the trimmed genotype matrix, using the EIGENSTRAT software. The first two principal components are displayed in two-dimensional space. Second, we performed UMAP clustering of the genotype matrix using the umap package for python software (*n* components = 2, default parameters). Third, we applied the UMAP method to the first 50 principal components of the trimmed genotypes used for the PCA-UMAP (*n* components = 2 and default parameters) [25,26]. To clarify the relationship between the position of each individual in the different downscaling methods, the software Grimon (graphical interface to visualize multi-omics networks; https://github.com/mkanai/grimon, accessed on 15 January 2022)) was used to visualize the two-dimensional graph, which was visualized as three-dimensional (3D) parallel coordinates and colored according to the color of the shin [27].

### 2.5. Whole-Genome Association Analysis and Linkage Disequilibrium Analysis

The GWAS analysis for shin color employed the linear mixed model of Effective Mixed Model Association eXpedited (EMMAX) software. The EMMAX software simplifies and saves the assumption that the effect of any given SNP on the trait is typically small; therefore, it only estimates the model variance components once per analysis to account for the population structure. The EMMAX estimates the variance components using the REML model:y=Xa+Zb+e

Where *y* is a vector of shin color; *X* is the incidence matrix for a random additive effect; *a* is the column vector of random additive effects; *Z* is the genotype value of the candidate SNP; *b* is the regression coefficient of the candidate SNP; and *e* is a random residual. The phenotypic variance–covariance matrix is var (*y*) = Var (*a*) + var (*e*) = *K σ_a_*^2^ + *I σ_e_*^2^, where *K* is the IBS kinship matrix; *I* is the identity matrix; *σ_a_^2^* is the additive variance; and *σ_e_^2^* is the variance of the random residuals.

### 2.6. Gene Ontology (GO) and Kyoto Encyclopedia of Genes and Genomes (KEGG) Analyses

Based on the LD attenuation distance calculated by PopLDdecay, the related genes in a certain region upstream and downstream of the physical location of the significant SNPs were annotated [28]. The sequences of the relevant genes were extracted from the mallard genome and translated into a protein sequence, which was then submitted to the KOBAS 3.0 server [29]. Chickens were selected as the reference species, and hypergeometric tests and Fisher’s exact test were used as the statistical methods.

### 2.7. qRT-PCR

Total RNA was extracted using the RNA Simple Total RNA Kit (Tiangen, Beijing, China), according to the manufacturer’s protocol, and 1 μg of total RNA was reverse-transcribed to cDNA, using the Fast Quant RT Kit (with gDNase) (Tiangen, Beijing, China). The gene expression was quantitatively analyzed by RT-qPCR in a QuantStudio™ 5 Real-Time PCR System, using the AceQ qPCR SYBR Green Master Mix (Low ROX Premixed; Vazyme, Nanjing, China), gene-specific primers (Table 1), and cDNA as a template. The cycling conditions were as follows: 95 °C for 5 min; followed by 40 cycles of 95 °C for 10 s; and 60 °C for 30 s. The Ct values were obtained using the default settings, and the relative mRNA expression of the target genes was calculated using the ^−2ΔΔ^Ct method after normalization to the levels of GAPDH (a constitutively expressed gene that was used as the internal control) [30].

### 2.8. Statistical Analysis

All of the results displayed in the bar graphs are expressed as the mean ± standard error of the mean (SEM) of three independent experiments. The statistical significance was determined using the student’s *t*-test in the stats package of R. The statistical significance was set at *p* < 0.05.

## 3. Results

### 3.1. Phenotypic Description and Population Structure Analysis

The ggcorr package of R and the GGally package were used to identify the correlation between shin color and sex. The results showed no correlations between the black, yellow, or spotted shin color and sex (Figure 1).

To reveal the finest-scale structure within the F_2_ population, we initially applied four dimensionality reduction methods to the large-scale F_2_ population genotype data: (1) PCA, an orthogonal linear transformation that projects the genotype data into a new reduced dimensional space so that larger variances are ordered (Figure 2a); (2) UMAP, a novel nonlinear dimensionality reduction technique, based on Riemannian geometry and algebraic topology, to model and preserve the high-dimensional topology of the data points in low-dimensional spaces (Figure 2b); (3) PC+UMAP, an application of UMAP for the main components of genotype data that is computationally more favorable, accurate, and statistically less noisy (PC+UMAP) (Figure 2c); and (4) MDS, a popular method for reducing dimensionality by obtaining the original sample set and computing a dissimilarity (distance) measure for each pairwise comparison of the samples. The samples were typically represented as two-dimensional graphs, such that the distances between the points on the graph were as close as possible to their multivariate dissimilarities (Figure 2d). Among the four methods of dimensionality reduction, we did not identify any individual separations caused by tibia color.

### 3.2. Genome-Wide Association Study Identified Candidate Variants of Shin Color

In this study, the GWAS was performed using EMMAX software. The Q–Q plots indicated that the model used for the GWAS analysis is reasonable. The λ (values (λ)) for the three different colored shins were 0.926 (black shin), 0.985 (spotted shin), and 0.878 (yellow shin), and the points in the upper-right corner of the Q–Q plot represented significant markers associated with the traits under study (Figure 3). Thus, the population stratification was adequately controlled. However, no significantly correlated loci for the GWAS analysis of spotted shins were found in the Q–Q plots, although a large number of potentially correlated loci for spotted shins were observed in the Manhattan plots.

For the spotted shins, six significant SNPs were identified, using a significant *p*-value ≤ 3.94885 × 10^−9^ as the threshold, while one extremely significant SNPs was identified using a *p*-value ≤ 7.8977 × 10^−10^ as the threshold. Most of these SNPs were located on chromosome 14 (ALP 14) (Figure 4, top). For the black shins, the Manhattan plot showed that a total of 2947 significant SNPs were identified, using a significant *p*-value ≤ 3.94885 × 10^−9^ as the threshold, while 2214 extremely significant SNPs were identified using a significant *p*-value ≤ 7.8977 × 10^−10^ as the threshold. Most of these SNPs were located on ALP 14 (Figure 4, middle). For the yellow shins, 4811 significant SNPs were associated with the spotted shins (*p*-value ≤ 3.94885 × 10^−9^) and 3451 extremely significant SNPs were identified, using a significant *p*-value ≤ 7.8977 × 10^−10^ as the threshold. Most of these SNPs were located on ALP 11 (Figure 4, bottom). We found that all six of the significant spotted shin-associated SNPs were shared with the black shins, based on a Venn analysis (Figure 5).

### 3.3. Functional Analysis of Shin Color Candidate Genes

To interpret the effect of the candidate loci on shin color, a GWAS was complemented with a genomic analysis, using the GO and the KEGG databases, which allowed for the further interpretation of the candidate locus functions to detect functional classes potentially involved in the shin color candidate loci. By annotating the candidate loci for black shins, we detected 80 genes, including *SLC25A43*, *SLC25A5*, *BMP15*, *SPRY3*, *POU4F3*, *EDNRB2*, and others associated with SNPs significantly associated with black shins (Appendix A). The GO, KEGG, and reactome enrichment analyses revealed that these genes were mainly involved in melanogenesis, necroptosis, calcium signaling, FoxO signaling, primary bile acid biosynthesis, apoptosis, cAMP signaling, MAPK signaling, JAK-STAT signaling, and other pathways (Figure 6a), as well as melatonin receptor activity, oxidative phosphorylation uncoupler activity, melatonin-binding interleukin-22 binding, interleukin-22 receptor activity, and endothelin receptor activity (Figure 6b). In addition, the SLC25A5,6 dimers exchanged ATP for ADP across the mitochondrial inner membrane (R-GGA-163215); mitochondrial uncoupling proteins (R-GGA-166187), ACOX2:FAD, and ACOXL:FAD oxidized (2S)-pristanoyl-CoA to trans-2,3-dehydropristanoyl-CoA dehydropristanoyl-CoA (R-GGA-389889); and other reactome pathways were significantly enriched (Figure 6c). For the yellow shins, we identified 53 candidate genes which included *FAM107A*, *FAM19A1*, *FAM19A4*, *FOXP1*, *SLC16A7*, *SLC25A20*, *SLC25A26*, *SYNPR*, and *MITF*, which were identified along with the other yellow shin-associated SNPs (Appendix A). The GO, KEGG, and reactome enrichment analyses revealed that these genes were mainly involved in melanogenesis, citrate cycle (TCA cycle), other types of O-glycan biosynthesis, glioma, primary bile acid biosynthesis, selenocompound metabolism, aldosterone synthesis and secretion, ECM-receptor interaction, and the Wnt signaling pathway (Figure 6d), as well as the terms laminin complex, ligase activity, protein disulfide oxidoreductase activity, S-adenosyl-L-methionine transmembrane transporter activity, and IMP dehydrogenase activity (Figure 6e). Meanwhile, the laminins bind HSPG2 (R-GGA-4084505); integrin alpha3beta1 and alpha6beta4 bind laminin-332, 511, 521, and (211, 221) (R-GGA-216048); integrin alpha7beta1 binds laminin-211, 221, 411, 512, and 521 (R-GGA-216058); integrin alpha6beta1 binds laminin-322, 512, 521, 211, 221, and 411 (R-GGA-3907292); USP3 and SAGA deubiquitinate histone H2A and H2B (R-GGA-5690080); laminins and nidogens bind collagen type IV networks (R-GGA-2426450); and laminin interactions were observed (R-GGA-3000157). The reaction pathway is also enriched (Figure 6f). For the spotted shins, the three genes *TMLHE*, *SPRY3* and *POU4F3* were associated with spot shin color (Appendix A).

### 3.4. EDNRB2 and MITF Isoform Expression Level in Black and Yellow Shin

Based on the GWAS analysis, we found that *MITF* and *EDNRB2* were candidate genes for the yellow and black shins. Therefore, *MITF* and *EDNRB2* may regulate pigmentation in duck shins. In addition, *MITF* consists of two isoforms, MITF-B and MITF-M, and the isoform-specific first exons are termed 1B and 1M in ducks, respectively (Figure 7a). Thus, we determined the expression levels of these two isoforms in the black and yellow shins. The results showed that MITF_12 was significantly expressed in the black shins (Figure 7b) and the MITF-M isoform was highly expressed in the yellow shins. The *EDNRB2* expression levels were compared between the black and yellow shins. The *EDNRB2* was significantly expressed in the black shins (Figure 7b). We also identified the genes downstream of the melanogenesis pathway (*TYR* and *TYRP1*). The tyrosinase production is regulated by *TYR* and *TYRP1*. The first stage of melanin formation is accomplished by tyrosinase, which changes the amino acid tyrosine, a component of proteins, into dopaquinone. The dopaquinone is then converted to melanin in the skin, hair follicles, iris, and retina by a sequence of chemical processes. The *TYR* and *TYRP1* expression showed that *TYR* and *TYRP1* were significantly expressed in the black shins (Figure 7b).

## 4. Discussion

While most of the studies on color deposition in animals and birds have focused on hair and feathers, skin color is equally important for animals and birds. The tibia is an important area of exposed skin in birds, and the color deposition has a direct impact on the effects of UV light. Although the feathers cover most of the skin of birds, the skin color plays a crucial role in birds because the deposition of melanin in the skin effectively reduces DNA damage caused by UV rays [31,32]. Diverse colors are observed in the skin of birds, and the skin of the tibia is mostly composed of black and yellow, although some spotted patterns composed of black and yellow are also present. The color of the shins, which is one of the few skin structures of birds exposed to the environment, is mainly determined by the concentration of melanin and carotenoids. Melanin is one of the main color pigments and a complex polymer, mainly derived from tyrosine [33,34]. The carotenoids, such as alpha-carotene and beta-carotene, are hydrocarbons, while the lutein-like compounds, such as lutein, zeaxanthin, and astaxanthin, contain oxygen atoms, such as hydroxyl groups [35]. The presence of carotenoids in feathers, skin, and muscles results in the carotenoid-based color [36]. In duck shins, the melanin deposition increases with age and UV exposure [37,38]. However, under the influence of domestication and selection, many duck breeds have shown a clearly fixed and stable inheritance of black shins, such as the CC duck, and most ducks that exhibit a black and yellow shin. However, some duck breeds also exhibit spotted shins. To identify the candidate genes that control shin color, the present study designed an F_2_ population cross between CC and CV ducks. In the black and yellow shins, we found significant signals and potential signals, while in the spotted shins, we identified two potential signals. In the black shins, 91 genes were annotated, including *EDNRB2*, whereas in the yellow shins, 93 candidate genes were annotated, including *MITF*. However, in the spotted shins, only three genes were annotated, namely, *TMLHE*, *SPRY3*, and *POU4F3*.

*EDNRB2* is a crucial gene for melanin production and a G protein-coupled receptor (GPCR) with seven transmembrane structure domains; moreover, it is a homolog of the endothelin receptor B (EDNRB), a gene involved in the formation of the melanocyte lineage [39]. The leukoplakia phenotype is caused by abnormalities in the endothelin and *EDNRB2* genes, which affect the ability of specific skin cells to synthesize pigments [40]. The aberrant proliferation, differentiation, survival, and migration of melanocytes and their progenitor cells, neural crest cells, cause this pattern [41,42]. The absence of neural crest-derived melanocytes results in a white coat in mice that exhibit homozygous lethal (s) mutations [41]. Previous studies have reported that *EDNRB2* is associated with white plumage phenotypes in Japanese quail, chickens, and domestic ducks [43,44]. Our findings also showed that EDNRB2 may control melanin. 

The *MITF* gene is a bHLHZip transcription factor belonging to the MYC superfamily [45,46]. *MITF* plays an important role in melanocytes and is a key regulator of melanin synthesis, and it is involved in the proliferation, differentiation, and transport of melanocytes [47]. Moreover, this gene has been shown to be involved in the regulation of many different events, including proliferation, differentiation, apoptosis, DNA replication, mitosis, and genomic stability [46]. Following the *MITF* silencing in human melanocytes, the expression of the *TYR* and *TYRP1* proteins was significantly reduced at different levels, while the expression of the *TYRP2* protein was significantly increased [48]. Tyrosinase (*TYR*) is involved in the synthesis of dopa and dopaquinone during melanin synthesis [49]. The type and amount of melanin synthesis depends on its activity and concentration [50]. The tyrosinase-related protein (*TYRP1*) gene is a member of the tyrosinase-related protein family and represents a key factor in melanin synthesis [49].

The yellow color of the skin, beak, and feet of most birds is caused by carotenoid deposition. The results showed that most members of the POU family are clearly associated with yellow tibiae, which have typical POU structural domains [51]. The POU family is a transcription factor family member that promotes the transcription of many genes related to development and metabolism, especially in Sherwang cell and progenitor cell development [52,53]. The POU transcription factors are distributed within the promoter region of *MITF* [38]. Therefore, we hypothesized that the yellow color of the tibia is co-regulated by the POU family and *MITF*; however, further experimental validation of the exact mechanism is needed. The present study provides new insights into the genetic factors that may influence the color diversity of birds’ tibiae; however, further experimental studies are needed to strengthen the hypotheses.

## 5. Conclusions

This study identified the candidate genes that are closely related to duck shin color. *MITF* and *EDNRB2* are candidate genes associated with shin melanosis. In addition, *POU4F3* is a candidate gene associated with shin color. We speculate that pigmentation may be coregulated by the POU family, *MITF*, and *EDNRB2* in the shins. However, the specific underlying mechanisms require further experimental verification.

## Figures and Tables

**Figure 1 genes-13-01391-f001:**
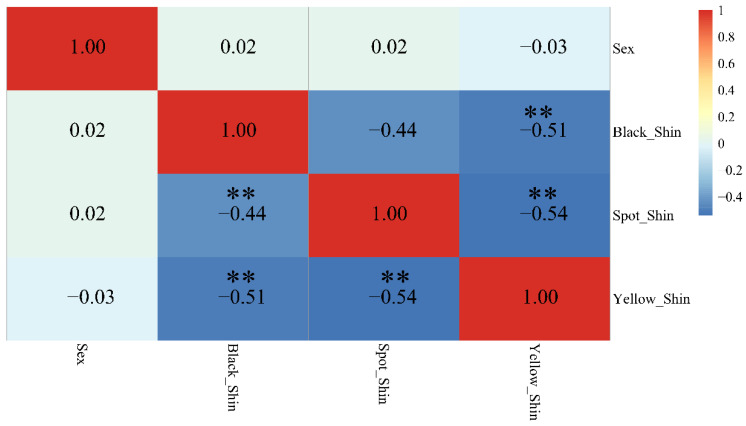
Correlation analysis of shin color and sex. ** represents *p* value ≤ 0.01.

**Figure 2 genes-13-01391-f002:**
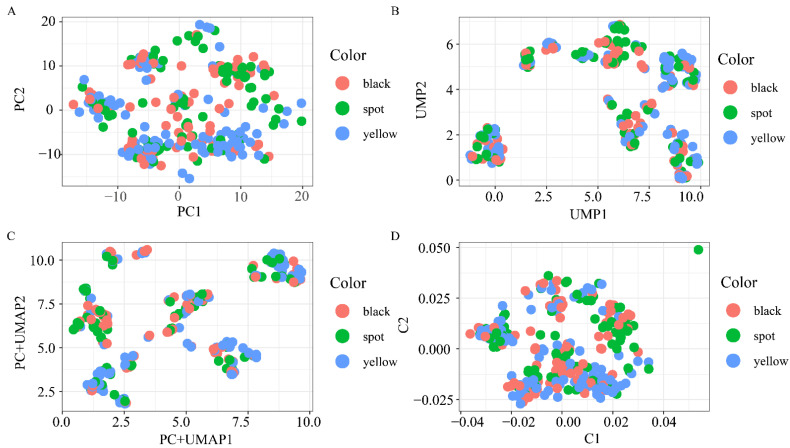
Two-dimensional illustrations of biobank-scale genotype data from the F_2_ population by the four dimensionality reduction methods: (**A**) PCA; (**B**) UMAP; (**C**) PC+UMAP; and (**D**) MDS. The color of individual dots indicates different shin colors.

**Figure 3 genes-13-01391-f003:**
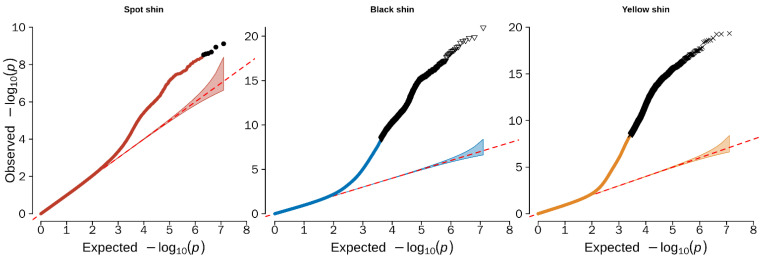
Quantile–quantile (Q–Q) plots of the GWAS for the shin color trait in ducks. The Q–Q plots show the late separation between the observed and expected values. The red dotted lines indicate the null hypothesis of no true association. Deviation from the expected *p*-value distribution is evident only in the tail area for each color shin, indicating that population stratification was properly controlled.

**Figure 4 genes-13-01391-f004:**
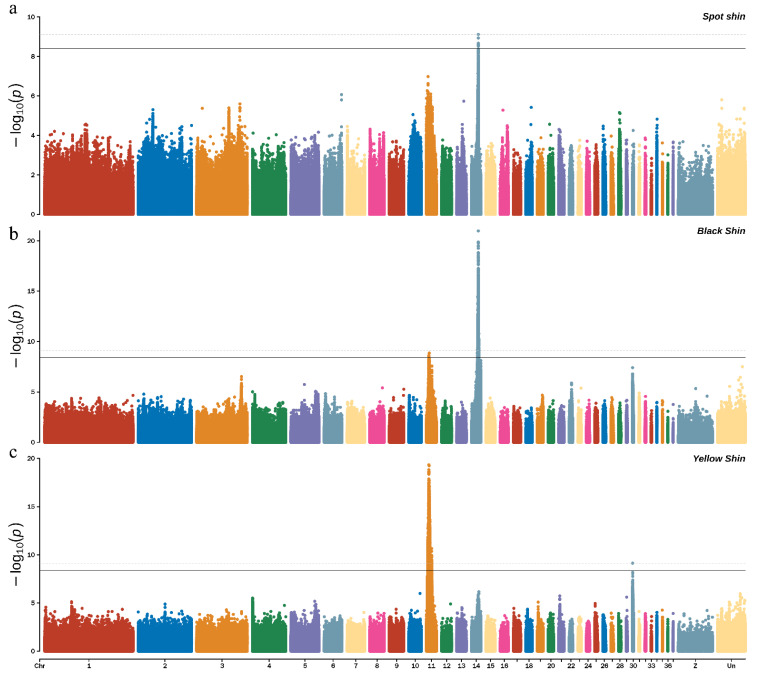
Manhattan plot shows the significance SNPs on the spot (**a**); black (**b**); and yellow (**c**) shin color by GWAS. The dotted line means threshold *p*-value = −log10 (7.8977 × 10^−10^) = 9.102; the black line means threshold *p*-value = −log10 (3.94885 × 10^−9^) = 8.403.

**Figure 5 genes-13-01391-f005:**
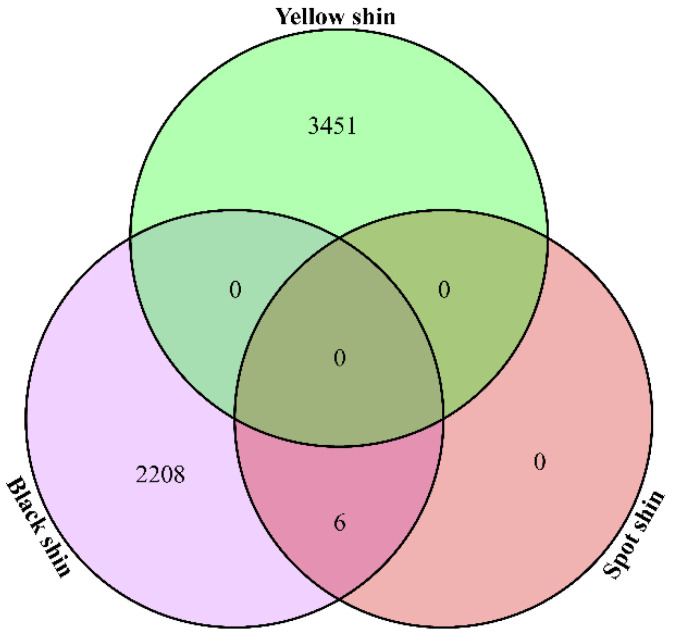
Venn analysis of all shin colors associated with significant SNPs.

**Figure 6 genes-13-01391-f006:**
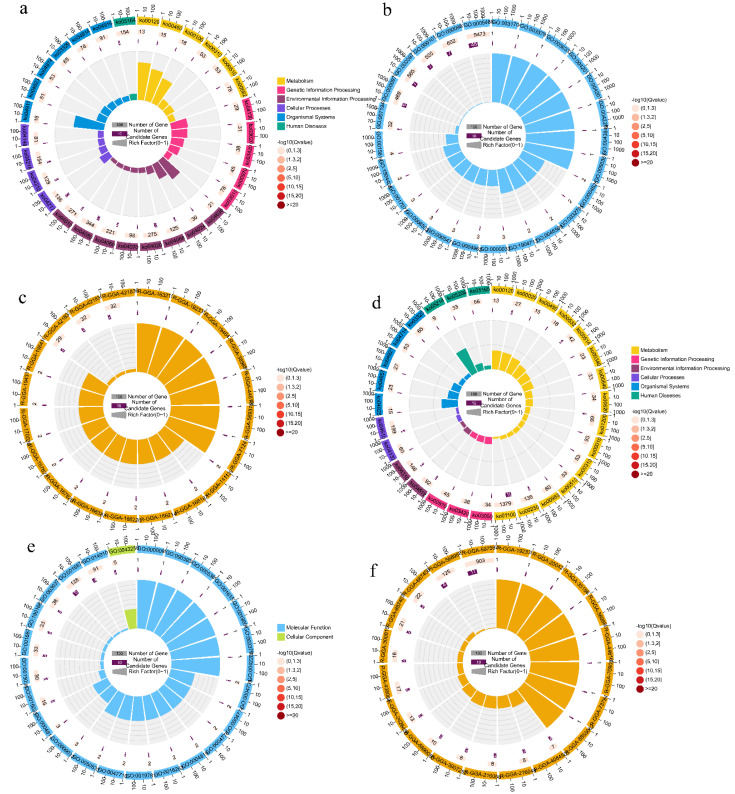
Functional enrichment analysis of the shin color candidate genes. (**a**–**c**) KEGG, GO, and Reactome pathway enrichment of black shin candidate genes; (**d**–**f**) KEGG, GO, and Reactome pathway enrichment of yellow shin candidate genes.

**Figure 7 genes-13-01391-f007:**
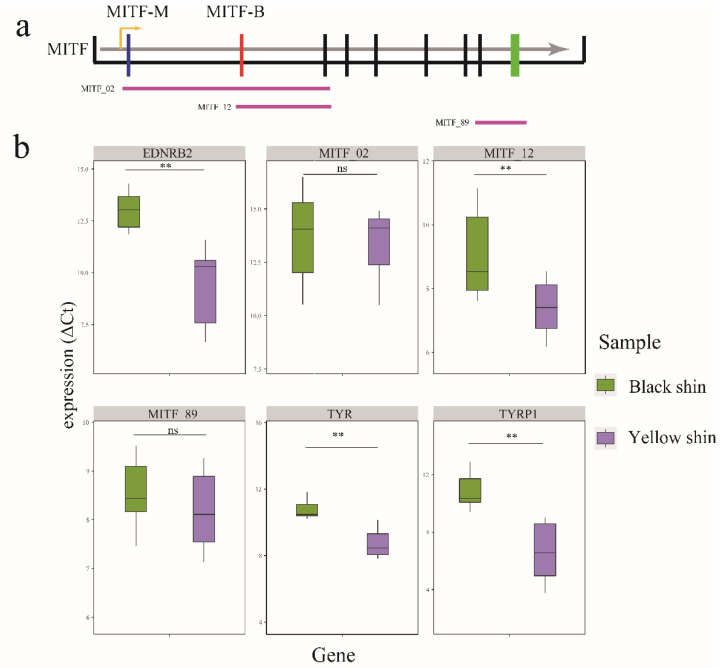
Expression differences of EDNRB2, TYR, TYRP1, and MITF in three exon junctions between black and yellow shins according to RT-qPCR. (**a**) MITF isoform. Exon 1M is specific for the MITF-M transcript, while exon 1B is specific for the MITF-B transcript; (**b**) EDNRB2, TYR, TYRP1, and MITF in three exon junctions between black and yellow shins. ** represents *p* value ≤ 0.01.

**Table 1 genes-13-01391-t001:** Primers used in the experiment in the present study.

Primers	Sequence (5′-3′)
*GAPDH*-F	GGTTGTCTCCTGCGACTTCA
*GAPDH*-R	TCCTTGGATGCCATGTGGAC
*MITF*-exon12-F	GCCAGACACCTGCCATCAAC
*MITF*-exon12-R	CTGCTTTACCTGCTGCCGC
*MITF*-exon02-F	TATGTGAATCGCTCAGACTGGAG
*MITF*-exon02-R	TGGTTGGCGTGTTTATTTGCTA
*MITF*-exon89-F	AACAGCAACGCACAAAGGA
*MITF*-exon89-R	GGTGGATGGCACAAGGGAC
*EDNRB2*-F	TGTAGAAGATGCCGGTGCATAC
*EDNRB2*-R	GAAGCCATAGCCTTTGACATGG
*TYR*-F	GGCAGACATCCAACTAACCCTA
*TYR*-R	GTCATTGTTCCCAGGATTTCGC
*TYRP*-F	TACAACATGGTGCCTTTTTGGC
*TYRP*-R	CATGCAGCAGCAGCAAAGATAA

## Data Availability

The genome assembly and all resequencing data used in this research are deposited in the Genome Sequence Archive (GSA) at the National Genomics Data Center (http://bigd.big.ac.cn/, (accessed on 17 July 2021)) Beijing Institute of Genomics, Chinese Academy of Sciences (GSA: CRA005019).

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
