# Peer review of "Genome-Wide Association Study for Screening and Identifying Potential Shin Color Loci in Ducks"

_genes, 2022, doi:10.3390/genes13081391_

Round 1

Reviewer 1 Report

The topic of the paper is very interesting and the authors have received lots of results, which they have extensively presented.  I only miss a few explanations:
1) Why did the authors choose to study breeds: Cherry Valley and Runju crested white ducks? Are chosen brees more interesting, more useful than another one?
2) Figure 1 can be better explained. It should contain a better legend, in presented form this figure can be not completely obvious
3) References should contain ";" between the next authors of the same reference.

Author Response

The topic of the paper is very interesting and the authors have received lots of results, which they have extensively presented. 

Response: Thank you for your kindly comment. We have revised the manuscript and made a point-by-point response to your comments. Please see below for further details.

I only miss a few explanations:
1) Why did the authors choose to study breeds: Cherry Valley and Runju crested white ducks? Are chosen brees more interesting, more useful than another one?

Response: Thank you for your kindly comment. The Cherry Valley Duck is a large bodied yellow-billed, yellow-tipped and white-feathered meat duck. The Chinese crested duck, on the other hand, is a small-bodied egg-laying duck with a black bill, black shins and white plumage. The two breeds were chosen because they are genetically distant and the phenotypic variation produced by the cross is relatively large, which helps to resolve many of the phenotypic candidate loci.

2) Figure 1 can be better explained. It should contain a better legend, in presented form this figure can be not completely obvious

Response: Thank you for your kindly comment. We have made changes to Figure 1 to better represent our results.

3) References should contain ";" between the next authors of the same reference.

Response: Thank you for your kindly comment. We have changed the mistake of reference.

Reviewer 2 Report

Genes

Genome-wide association study for screening and identifying potential shin color loci in ducks

This study entitled ‘Genome-wide association study of screening and identifying potential shin color loci in ducks’ is interesting to the reader. However, there need some corrections. Please see below comments.

Major revision

1.     I felt that this study needs to be clearer and change all of the figures. Because the quality of some figures is too low to be review.

2.     It is better to mention how you will use these results at the end of conclusion or abstract.

3.     How about add the raw data at supplementary file? I have seen or reviewed papers with the same analysis in duck. When I received this paper, it felt like the authors using the same data.

4.     The authors have to check the primers and add more information like accession number or other identifications. MITE-exon02, EDNRB2, and TYR are not detected in ‘Primer Blast’. Authors can check in ‘https://www.ncbi.nlm.nih.gov/tools/primer-blast/’.

Minor revision

1.     Some sentences require the addition of a references. L39-41, L152-155

2.     In materials and methods, need to write the sex ratio or number of ducks used in the experiment.

3.     L133. ‘?e2  ??2

4.     Figure 1. The shape of the parentheses shown in the figure 1 is different. This should be checked and corrected. If it expressed ‘<’ or ‘’, an explanation should be written in the footnote of the figure.

5.     L194. Footnote is different with L177-182.

6.     L210. ‘The red lines….’ It need to change as ‘The red dotted lines…’ etc.

7.     Figure 4. It looks better to express each figures as a, b, or c.

8.     Figure 6. I couldn’t see this figure clearly, and even read the letters or numbers.

9.     Figure 7 footnote. ‘b) EDNRB2, TYR…’ comma is missing. And need to explain the statistical indicators.

Author Response

This study entitled ‘Genome-wide association study of screening and identifying potential shin color loci in ducks’ is interesting to the reader. However, there need some corrections. Please see below comments.

Response: Thank you for your suggestion. We have revised the manuscript and made a point-by-point response to your comments. Please see below for further details.

Major revision

  1. I felt that this study needs to be clearer and change all of the figures. Because the quality of some figures is too low to be review.

Response: Thank you for your suggestion. We have improved the clarity of the images in the revised version.

  1. It is better to mention how you will use these results at the end of conclusion or abstract.

Response: Thank you for your kindly suggestion. According to your suggestion, we have added a section on future applications of the screened candidate genes in the abstract and conclusion of the manuscript.

  1. How about add the raw data at supplementary file? I have seen or reviewed papers with the same analysis in duck. When I received this paper, it felt like the authors using the same data.

Response: Thank you for your kindly comment. We have added our phenotype file.

  1. The authors have to check the primers and add more information like accession number or other identifications. MITF-exon02, EDNRB2, and TYR are not detected in ‘Primer Blast’. Authors can check in ‘https://www.ncbi.nlm.nih.gov/tools/primer-blast/’.

Response: Thank you for your kindly comment. Based on your comment, we matched three primer pairs to the duck genome in NCBI and all primers were uniquely matched to the corresponding genes according to the results of the matching.

MITF-exon02

F:TATGTGAATCGCTCAGACTGGAG

R: TGGTTGGCGTGTTTATTTGCTA

EDNRB2

F:TGTAGAAGATGCCGGTGCATAC

R:GAAGCCATAGCCTTTGACATGG

TYR

F:GGCAGACATCCAACTAACCCTA

R:GTCATTGTTCCCAGGATTTCGC

Minor revision

  1. Some sentences require the addition of a references. L39-41, L152-155

Response: Thanks for your kindly comment. We have added the appropriate references in the corresponding paragraphs.

  1. In materials and methods, need to write the sex ratio or number of ducks used in the experiment.

Response: Thanks for your kindly comment. We have added the sex ratio or number of ducks used in the experiment into part of materials and methods.

  1. ‘?e2  → ??2

Response: Thanks for your kindly comment. We have changed this mistake.

  1. Figure 1. The shape of the parentheses shown in the figure 1 is different. This should be checked and corrected. If it expressed ‘<’ or ‘≤’, an explanation should be written in the footnote of the figure.

Response: Thanks for your kindly comment. We have changed the figure 1.

  1. Footnote is different with L177-182.
  2. L210. ‘The red lines….’ It need to change as ‘The red dotted lines…’ etc.
  3. Figure 4. It looks better to express each figures as a, b, or c.
  4. Figure 6. I couldn’t see this figure clearly, and even read the letters or numbers.
  5. Figure 7 footnote. ‘b) EDNRB2, TYR…’ comma is missing. And need to explain the statistical indicators.

Response: Thanks for your kindly comment. We have changed the above mistake point by point. 

Round 2

Reviewer 2 Report

Re-submitted paper have been almost corrected according to the comments. However, there are some additional minor comments below. please check those things. 

1. Reference format should be modified according to the Genes style.

2. Signs of significance in Figure 1 should be consistent.

(ex. display only in the upper right, or only in the lower left, or all.)

3. Figure 6. is still difficult to recognize.